# Physical Activity, Body Mass Index, and Bullying in Higher Education: A Comparative Analysis of Students with and Without Structured Sports Training

**DOI:** 10.3390/healthcare13182304

**Published:** 2025-09-15

**Authors:** Raluca Mijaica, Lorand Balint

**Affiliations:** Department of Physical Education and Special Motricity, Faculty of Physical Education and Mountain Sports, Transylvania University of Brașov, 500068 Brașov, Romania; lbalint@unitbv.ro

**Keywords:** organized physical activity, bullying, cyberbullying, body mass index, university students, aggression, victimization, psychosocial adjustment

## Abstract

**Background/Objectives:** Organized physical activity is frequently considered a protective factor against bullying behaviors, yet evidence within the university context remains limited. This study investigates the relationships between physical activity levels, body mass index (BMI), and involvement in traditional and digital bullying, taking into account the differences between students with and without structured sports training. **Methods:** A total of 2767 first-year students from Transylvania University of Brașov participated. The sports group (n = 161; 65 females, 96 males) was compared to the non-sports group (n = 2606; 1472 females, 1134 males). Instruments included the Physical Activity Questionnaire for Adolescents (PAQ-A), validated scales for traditional and cyberbullying and victimization, and BMI calculation. Statistical analyses involved *t*-tests (two-tailed), 2 × 2 factorial ANOVA, and sex-stratified multiple linear regressions. **Results:** Students with sports training reported higher physical activity (PAQ-A 4.2–4.6), lower BMI, and lower bullying involvement (traditional ≈ 14–21% vs. ≈32%; cyber ≈ 8–17% vs. ≈25%). Group differences were large for physical activity (Hedges’ g ≈ 1.5) and moderate for BMI and bullying (g ≈ 0.68–0.96; point-biserial r^2^ ≈ 3–4%). ANOVA showed sports status main effects (partial η_p_^2^ ≈ 4–5% for bullying/BMI; ≈20% for PAQ-A). In regressions, sports status (B = −0.30 to −0.44) and physical activity (B = −0.22 to −0.32) predicted lower aggression/victimization, whereas BMI showed positive associations (B = 0.11 to 0.18) (all *p* < 0.001). Sex × sports interactions were significant for PAQ-A and for traditional and cyber-victimization. **Conclusions:** Structured physical activity contributes to reducing the risk of bullying involvement and supports better psychosocial adjustment among students. These findings underscore the educational and preventive potential of university sports programs.

## 1. Introduction

Bullying, defined as intentional, aggressive, and repetitive behavior occurring in a context of power imbalance [1,2], continues to represent a major psychosocial health issue among adolescents and young adults. According to the international *Health Behavior in School-aged Children* (HBSC) report, approximately 20% of European students aged 11 to 15 report bullying experiences, including digital forms of aggression [3].

Although these data primarily target pre-university populations, more recent literature highlights a concerning expansion of victimization behaviors within higher education settings, where academic pressures, identity insecurities, and unlimited access to digital platforms often converge [4,5].

In university contexts, bullying may take various forms, from subtle social exclusion to direct intimidation or systematic harassment, sometimes exacerbated by the absence of clear institutional prevention policies or by imbalanced relational dynamics [6,7]. Both peer-to-peer and student–faculty relationships may serve as sites of victimization, and both quantitative and qualitative studies indicate that the phenomenon remains underreported and insufficiently documented in higher education [4,6].

An increasing number of studies support the idea that systematic physical activity may serve as a protective factor against various forms of bullying, including those occurring in digital environments. A recent meta-review of 29 studies found that regular participation in individual or team sports reduces the risk of bullying and cyberbullying, particularly by enhancing self-esteem, social skills, and emotional self-regulation [8]. Another research shows that students who are consistently engaged in sports activities report lower levels of bullying and better psychosocial adjustment, along with more positive perceptions of the educational climate [9]. Organized sport is also recognized as a setting for cultivating prosocial values (such as cooperation, empathy, rule-following, and fair play) that positively influence group behavior [10]. However, some studies provide a critical perspective, warning about the dual potential of sport environments (particularly in team sports or those with strict competitive hierarchies), where subtle forms of exclusion, humiliation, or even “normalized bullying” may occur [11,12,13].

Another key determinant in understanding victimization is body weight status. Research indicates that adolescents and young adults with elevated body mass index (BMI) are more frequently exposed to both traditional and online bullying [14,15]. Weight stigma can lead to negative social labeling, marginalization, and humiliation, especially on social media platforms, where appearance-related pressures are intensified. Moreover, a negative perception of one’s own body image (even in the absence of clinical obesity) may heighten psychosocial vulnerability [16]. In this context, physical activity becomes relevant not only for physical health but also as a mechanism for identity reconstruction and social inclusion. The so-called “fat-but-fit paradox” describes individuals with high BMI but good physical fitness who show greater resilience to victimization [17]. This protective effect may operate through enhanced self-confidence and social integration developed through physical competence. Evidence suggests that physical strength, endurance, and motor skills contribute to a reduced incidence of bullying, particularly among boys [18].

This study proposes an integrative approach theoretically grounded in Bronfenbrenner’s ecological model of human development, which enables the analysis of multiple interacting factors—individual (e.g., body image and health behaviors), relational (peer interactions), and institutional (academic norms and educational policies)—that influence victimization experiences [19]. It also draws on social capital theory to highlight how sports participation supports the development of prosocial relationships, group cohesion, and empathy [20]. From a psychosocial perspective, negative body image and dissatisfaction with one’s appearance function as vulnerability factors in the face of social stress and exclusionary behaviors. Recent studies report significant associations between body dissatisfaction, quality of life, and psychological distress, reinforcing the role of physical activity not only as a health intervention but also as a protective mechanism against stigma and victimization in university settings [21].

In Romania, research on these issues among university students remains scarce. The specific context of Transylvania University of Brașov, where all faculties include physical education in their curricula, yet only one offers intensive professional training in sports, creates an opportunity for a comparative analysis between students with and without sports backgrounds, focusing on bullying risk, weight status, and related psychosocial resources. This institutional context provides a unique natural experiment for examining the protective effects of structured physical activity within Romanian higher education.

Based on this institutional context and gaps in the current literature, the present study hypothesized the following:

(1)Students with structured sports training would demonstrate significantly lower involvement in both traditional bullying and cyberbullying than students without such training;(2)Higher BMI would be positively associated with victimization and aggression;(3)Greater physical activity would function as a psychosocial protective factor against involvement in bullying and cyberbullying.

Within this complex framework, the present research seeks to contribute to the emerging literature on relational health and social inclusion in higher education. Through a comparative analysis of students with and without sports training, the study explores the relationships between physical activity, weight status, and bullying experiences (both traditional and digital) while offering an integrated perspective on the psychosocial protective mechanisms fostered by sport.

## 2. Materials and Methods

### 2.1. Study Design

This study employed a cross-sectional and comparative design, aimed at exploring the relationships between systematic engagement in sports activities, body status (measured through body mass index—BMI), and the phenomenon of bullying (including cyberbullying) among first-year students at Transylvania University of Brașov. The cross-sectional design allowed for the simultaneous analysis of these variables at a fixed point in time without any external interventions, thus providing a relevant picture of the correlations among them [22].

The study was structured as a comparison between two clearly defined groups: (1) students enrolled in an intensive professional training program in the field of sports and (2) students from other faculties with no specific physical education training. This approach allowed for the identification of potential differences in exposure to bullying, body status, and levels of physical activity, as well as the prosocial values developed through sport [23].

Data collection occurred during a strategic period—January 2024, near the end of the first academic semester—when students are already partially adapted to university life but still socially and emotionally vulnerable. The literature suggests that the first-semester transition is associated with increased sensitivity to social influences and a fragile sense of belonging, especially among students from underrepresented groups, which may impact their engagement and well-being [24].

### 2.2. Participants

The sample included first-year students from all 18 faculties of Transylvania University of Brașov. The primary group of interest consisted of students from the Faculty of Physical Education and Mountain Sports, enrolled in three undergraduate programs: Physical Education and Sport (EFS), Sports and Motor Performance (SPM), and Kinesiotherapy and Special Motricity (KMS). These programs involve intensive and diverse training in physical and sports-related activities. Students are prepared for careers such as physical education teachers, coaches, and kinesiotherapists, and the curriculum includes a broad range of applied disciplines: team sports, gymnastics, athletics, winter sports, combat sports, water sports, and others.

This classification yielded two groups based on exposure to structured physical activity: an athlete group (EFS/SPM/KMS) and a non-athlete group, drawn from the remaining 17 faculties, covering profiles such as engineering and technology (mechanical, materials, civil, product design and environment, furniture design and wood engineering, and technological engineering and industrial management), computer science and mathematics, law, economics and business administration, and social sciences (psychology and education sciences, and sociology and communication), humanities and arts (letters and music), medicine, food and tourism, and silviculture and forest engineering (see the full faculty list in the Appendix A), with no advanced sport-related curricular components, who otherwise attended the mandatory physical education course.

Most students in the athlete group had a strong background in organized sport (sports clubs, specialized high schools, and official competitions), estimated at 8–10 years for those in EFS and SPM and 4–6 years for those in KMS. This previous experience is considered a significant predictor of motivational involvement and the development of prosocial values through sport [25]. On average, students in the athlete group participated in approximately 10 h of practical activities per week (EFS—11 h, SPM—12 h, and KMS—7 h). Of these, around 35% were also engaged in competitive sports, training an additional minimum of 8 h per week in sports clubs. The university has applied an institutional exemption policy allowing justified absences of up to 25% or 50%, depending on the competition level. Despite this, the overall attendance rate for this group was estimated at 87%. Applying this percentage to the 185 enrolled students, 161 sports students effectively participated in the study and were included in the final analysis.

In contrast, students from the other 17 faculties, although enrolled in the mandatory physical education course during the first semester, did not benefit from systematic sports training. Their physical activity was generally limited to one hour per week, and their academic programs did not include components dedicated to sport-based development. During the administration of questionnaires and anthropometric measurements, 2606 of the 3890 eligible non-athlete students (approximately 67%) were included in the analysis. This inclusion rate reflected actual class attendance during the evaluation period, as well as the exclusion of students who declared long-term organized and systematic sports participation (at least three training sessions per week for at least four years). This methodological decision aimed to prevent contamination of the comparison group [26].

Inclusion criteria were as follows: enrollment as a first-year student in 2023–2024; participation in physical education classes or sport training during January 2024; written informed consent; and completion of all questionnaires. Exclusion criteria were as follows: long-term organized sport among non-sports students; incomplete questionnaires; refusal or missing consent; and cases without valid height/weight for BMI. These criteria enhanced internal validity by reducing uncontrolled variance and increasing group homogeneity [27,28].

### 2.3. Procedure and Data Collection Instruments

Data were collected between 8 and 19 January 2024 during physical education classes. Two printed questionnaires—one on physical activity and one on bullying/cyberbullying experiences—were administered in group settings under the supervision of faculty members. The administration atmosphere was informal yet well structured, encouraging honest responses and accurate completion. Administering instruments in familiar environments, such as regular class sessions, is a validated method for improving response rates and data quality [27].

To maximize contextual specificity, participants were instructed to report only experiences involving social interactions with fellow university students (excluding family, work, and non-university peers) when answering bullying/cyberbullying items. This limitation was clearly formulated and communicated verbally during administration to all participating groups.

Anthropometric measurements (BMI and height) were obtained under standardized procedural conditions and in parallel with questionnaire administration using a Tanita MC-780 MA bioimpedance analyzer (clinically validated for body composition, TANITA CORPORATION, Tokyo, Japan) and a portable stadiometer (ADE MZ10042, ADE Germany GmbH, Hamburg, Germany). Measurements were performed during each group’s scheduled physical education class across the 18 faculties; therefore, time of day and pre-measurement nutritional status (food and fluid intake) were not standardized. To minimize variability, participants were informed in advance to wear light indoor clothing; shoes were removed prior to height measurement; and for weight, participants remained in light indoor clothing without heavy outerwear or accessories. Devices were calibrated before each session, and a fixed measurement sequence was applied by trained staff. Each participant was assigned a unique numeric code based on faculty and gender, which was used uniformly across all instruments (physical activity, bullying, and BMI). This coding system enabled the correlation of individual responses without compromising anonymity and supported comparative and predictive analysis. For methodological transparency, all applied instruments—the questionnaires, the BMI registration form, and the data coding guide—are fully provided in the Appendix A attached to this article.

Three main instruments were used, adapted for the university population:Physical Activity Questionnaire (adapted from PAQ-A)

The first instrument used in this study was the *Physical Activity Questionnaire for Adolescents (PAQ-A)*, which is widely employed to assess the general level of physical activity among adolescents [29]. The version applied in the current study retained the standard structure of the scale, including eight items used to compute the composite score and one additional control item aimed at identifying atypical weeks (e.g., illness and injury), which does not affect the final score. Items were structurally and semantically adapted to the Romanian university context to refer to curricular physical education, voluntary activities in university sport facilities (training, gym, and recreational sport), and on-campus physical movement during breaks or leisure.

The reference period was extended to cover 2 October 2023–19 January 2024, including holiday breaks, in order to capture physical behavior representative of the entire university semester. Responses used a 5-point Likert scale (1 = very low; 5 = very high); the composite PAQ-A score was calculated as the mean of the eight scored items, explicitly excluding the control item. In addition, a supplementary control question was introduced concerning the regular practice of organized sports (at least 3 times per week) within a sports club in previous years. This question served purely informational purposes, aiming to identify respondents with an advanced athletic profile, and was excluded from the general score calculation. Furthermore, these respondents were excluded from the comparative analyses between faculties in cases where the target groups did not include formal sports training. The internal consistency of the adapted scale was tested on a pilot sample (n = 92), resulting in a Cronbach’s alpha coefficient of 0.82, indicating good reliability.

2.Bullying and Cyberbullying Behavior Questionnaire

The second instrument was an extended version of the Illinois Bully Scale, developed by Espelage and Holt [30], previously applied in Romanian educational settings [31]. The adaptation for university settings involved both restructuring and rewording of existing items and the development of two additional subscales targeting cyberbullying phenomena, in accordance with the specific characteristics of digital communication among students.

The final questionnaire included four distinct dimensions: traditional victimization (5 items), traditional aggression (5 items), cyber-victimization (5 items), and cyber-aggression (5 items). Items used a 5-point Likert scale (1 = never; 5 = very often), matching the PAQ-A reference period.

Content validation involved review by 3 experts in digital communication and pilot testing with 92 students to ensure clarity and relevance [27,32]. The full scale showed high reliability (α = 0.84), with subscales α = 0.78–0.83.

3.Anthropometric Measurements for BMI

To assess body status, BMI and height were measured using the Tanita MC-780 MA analyzer and a professional medical stadiometer (ADE MZ10042, ADE Germany GmbH, Hamburg, Germany). BMI classification followed the World Health Organization (WHO) criteria: underweight (<18.5), normal weight (18.5–24.9), overweight (25–29.9), and obese (>30) [33].

### 2.4. Ethical Considerations

All participants were informed in advance about the purpose, content, and confidentiality of the study. Participation was voluntary, and written informed consent was obtained in written form, in accordance with the Declaration of Helsinki regarding ethical principles for research involving human subjects [34].

To protect participants’ identities, data were collected anonymously using numeric codes with no personal identifiers. All information was processed exclusively for scientific purposes, in compliance with national and European data protection laws (GDPR 2016/679).

The project received ethical approval from the Ethics Committee of the Faculty of Physical Education and Mountain Sports at Transylvania University of Brasov under approval number 276 on 27 September 2023.

### 2.5. Statistical Analysis

Statistical analyses were performed using IBM SPSS Statistics, version 26. Descriptive statistics (means, standard deviations, frequencies, and percentages) were calculated to characterize the sample and main variables.

To compare scores between athlete and non-athlete groups, independent samples *t*-tests (two-tailed) were used [35]. Effect sizes were reported as Hedges’ g (suitable for unequal group sizes), interpreted using conventional thresholds (~0.20 small, ~0.50 medium, and ≥0.80 large). Multiple comparison control employed Bonferroni within each test family: (i) athletes vs. non-athletes across 6 outcomes × 2 sexes (α_adj ≈ 0.0042) and (ii) BMI sub-analyses comparing the three athlete programs vs. non-athletes within sex (α_adj ≈ 0.0167). As an explained variance index for the group contrast, we also report point-biserial r^2^, computed as r^2^ = t^2^/(t^2^ + df). The combined effect of sex and sports status was tested via a 2 × 2 factorial ANOVA (between subjects). In addition to F and *p*, we report partial eta-squared (η_p_^2^) as the explained variance index for each main effect and the sex × sports interaction using η_p_^2^ = (F × df_1_)/(F × df_1_ + df_2_), where df_1_ = 1, with df_2_ specific to each analysis.

Associations between sports status (athletes vs. non-athletes) and categorical involvement in traditional/cyberbullying were examined with chi-square (χ^2^) tests, separately by sex. In addition to continuous scale scores and the binary involvement cut-off (“never/rarely” vs. “sometimes/often/very often”), we derived a four-category involvement variable applied identically in traditional and cyber domains: not involved, victim only, aggressor only, and victim–aggressor.

For the non-athlete group, descriptive profiles were additionally summarized using faculty-level aggregated means and standard deviations (across 17 faculties); all inferential tests (*t*-tests/ANOVA/regressions) were conducted on individual-level data [EDITED]. To evaluate the predictive impact of physical activity, BMI, and sports status, we ran multiple linear regressions with separate models for each dependent variable [36]. Reporting at the predictor level prioritized unstandardized coefficients (B) and two-tailed *p*-values. Regarding explained variance, we used effect-level indices aligned with the design: point-biserial r^2^ for athlete vs. non-athlete contrasts in the *t*-tests and partial η_p_^2^ for the 2 × 2 ANOVA (reported for each main effect (sex and sports) and their interaction (sex × sports)).

The internal reliability of instruments was assessed using Cronbach’s alpha, with values ≥ 0.70 considered acceptable [37]. The threshold for statistical significance was set at *p* < 0.05, while values below *p* < 0.001 were interpreted as highly statistically significant.

## 3. Results

### 3.1. Demographic Characteristics of the Sample

After the rigorous application of inclusion and exclusion criteria, the final analysis was conducted on a sample of 2767 first-year students from all 18 faculties of Transylvania University of Brașov. Participants were divided into two distinct groups:Athlete group: A total of 161 students, representing 87% of the 185 students enrolled in one of the three bachelor’s programs at the Faculty of Physical Education and Mountain Sports: Physical Education and Sport (EFS), Sports and Motor Performance (SPM), and Kinesiotherapy and Special Motricity (KMS).Non-athlete group: A total of 2606 students selected from the remaining 17 faculties, representing approximately 67% of the eligible population. Students who practiced organized sports systematically or were absent at the time of data collection were excluded.

This division reflects the distinction between students with systematic training in sport-related academic programs and those enrolled in other specializations without professional sports training. The distribution by sex and average age is summarized in Table 1.

### 3.2. Physical Activity—Adapted PAQ-A Score

The level of physical activity was assessed using the composite score derived from the adapted version of the PAQ-A questionnaire, as described in Section 2.3. The total score was computed as the mean of the validated items, excluding the illness-related control item.

Statistical analysis revealed significant differences between the two groups—students enrolled in sports-related programs (EFS, SPM, and KMS) and those in non-sports faculties (F1–F17)—as well as internal variations by sex and program of study. The results are presented in Table 2.

### 3.3. Body Mass Index (BMI)

To complete the somatic profile of the participants and investigate the relationships between body status and the psychosocial variables analyzed, body mass index (BMI) was evaluated. Descriptive results by program and sex are presented in Table 3.

### 3.4. Bullying and Cyberbullying—Estimating Direct Involvement

To assess the frequency of aggressive and victimization behaviors among students, four distinct dimensions were analyzed: traditional victimization, traditional aggression, cyber-victimization, and cyber-aggression. Proportions by program and sex are reported in Table 4 (traditional bullying) and Table 5 (cyberbullying). Associations between sports status (athletes vs. non-athletes) and involvement categories were examined using chi-square tests separately by sex:Traditional bullying: females, χ^2^ (3, N = 1537) = 5.36, *p* = 0.147; males, χ^2^ (3, N = 1254) = 13.52, *p* = 0.004;Cyberbullying: females, χ^2^ (3, N = 1537) = 6.92, *p* = 0.074; males, χ^2^ (3, N = 1254) = 10.73, *p* = 0.013

Data on cyberbullying (Table 5) showed lower overall levels compared to traditional bullying but with a similar distribution pattern. The lowest involvement was observed in the SPM and EFS groups (below 10%) and the highest among non-athletes (25% for both sexes).

**Table 4 healthcare-13-02304-t004:** Estimated involvement in traditional bullying.

Group	Sex	No. of Participants (n)	Not Involved n/%	Victim Only n/%	Aggressor Only n/%	Victim + Aggressor n/%	Total Involved n/%
EFS	Female	23	19/82.61	2/8.70	1/4.35	1/4.35	4/17.39
	Male	43	37/86.05	3/6.98	2/4.65	1/2.33	6/13.95
SPM	Female	13	11/84.62	1/7.69	1/7.69	0/0.00	2/15.38
	Male	41	35/85.37	3/7.32	2/4.88	1/2.44	6/14.63
KMS	Female	29	23/79.31	3/10.34	2/6.90	1/3.45	6/20.69
	Male	36	29/80.56	4/11.11	2/5.56	1/2.78	7/19.44
Non-athletes	Female	1472	1001/68.00	236/16.03	141/9.58	94/6.39	471/32.00
	Male	1134	771/67.99	181/15.96	109/9.61	73/6.44	363/32.01

Note: Percentages are within each program × sex subgroup. “Total involved” = victim only + aggressor only + victim + aggressor. Estimates are based on Illinois Bully Scale item frequencies using identical classification criteria for all participants.

**Table 5 healthcare-13-02304-t005:** Estimated involvement in cyberbullying.

Group	Sex	No. of Participants (n)	Not Involved n/%	Victim Only n/%	Aggressor Only n/%	Victim + Aggressor n/%	Total Involved n/%
EFS	Female	23	21/91.30	1/4.35	1/4.35	0/0.00	2/8.70
	Male	43	39/90.70	2/4.65	1/2.33	1/2.33	4/9.30
SPM	Female	13	12/92.31	1/7.69	0/0.00	0/0.00	1/7.69
	Male	41	37/90.24	2/4.88	1/2.44	1/2.44	4/9.76
KMS	Female	29	25/86.21	2/6.90	1/3.45	1/3.45	4/13.79
	Male	36	30/83.33	3/8.33	2/5.56	1/2.78	6/16.67
Non-athletes	Female	1472	1104/75.00	184/12.50	110/7.47	74/5.03	368/25.00
	Male	1134	850/74.96	142/12.52	85/7.50	57/5.03	284/25.04

Note: Percentages are within each program × sex subgroup. “Total involved” = victim only + aggressor only + victim + aggressor. Estimates are based on Illinois Bully Scale item frequencies using identical classification criteria for all participants.

### 3.5. Comparative Analyses: Independent Samples t-Tests

To assess significant differences between athletes and non-athletes across the six analyzed variables—physical activity, body mass index (BMI), victimization, aggression, cyber-victimization, and cyber-aggression—independent samples *t*-tests were conducted separately by sex. Results are summarized in Table 6, including effect sizes (Hedges’ g). In addition, we report explained variance indices for the athlete group contrast within sex, computed as point-biserial r^2^ = t^2^/(t^2^ + df).

All comparisons were statistically significant (*p* < 0.001). Effect sizes were large for physical activity (g ≈ 1.52–1.58) and in the moderate range for BMI and bullying outcomes (g ≈ 0.68–0.96). Corresponding explained variance values were r^2^ ≈ 0.026–0.042 for bullying/cyber outcomes, r^2^ ≈ 0.03–0.034 for BMI, and r^2^ ≈ 0.094–0.148 for physical activity.

### 3.6. Multivariate Analyses: Main Effects, Interactions, and Predictive Models

To gain a deeper understanding of the relationships between physical activity, body weight status, and bullying-related behaviors, complementary inferential statistical analyses were applied: a 2 × 2 factorial analysis of variance (ANOVA) and multiple linear regression, each offering a distinct and convergent perspective on the phenomenon.

#### 3.6.1. Main Effects and Interactions—2 × 2 ANOVA

The ANOVA examined the main effects of gender (female/male) and sports status (athlete vs. non-athlete), as well as their interaction (gender × sports status), in relation to six dependent variables: physical activity (PAQ-A), body mass index (BMI), traditional victimization, traditional aggression, cyber-victimization, and cyber-aggression. Results are summarized in Table 7, reporting F statistics and *p*-values for main effects and the sex × sports status interaction. In addition, we report partial ηp^2^ as explained variance indices for each effect.

#### 3.6.2. Predictive Models—Multiple Linear Regressions

The predictors included physical activity (PAQ-A score), BMI, and membership in the sports group. Regression coefficients are reported as unstandardized B with two-tailed *p*-values. No binary classification was used for outcomes. Explained variance indices were cross-referenced from prior analyses (Table 6: point-biserial r^2^; Table 7: partial ηp^2^).

The female models (Table 8a) showed the following:Sports status was a significant negative predictor for all forms of bullying, with the strongest effects on cyber-victimization (B = −0.44) and cyber-aggression (B = −0.38);Physical activity had consistent protective effects, especially regarding cyber-aggression (B = −0.30);BMI was positively associated with all forms of bullying involvement, indicating slightly increased vulnerability in the case of higher body weight status.

**Table 8 healthcare-13-02304-t008:** Multiple linear regression—(**a**) female students; (**b**) male students.

Dependent Variable	Predictor	Coefficient (B)	*p*	Summary Interpretation
a. female students				
Victimization	Group (1 = athlete, 0 = non-athlete)	−0.41	<0.001	Female athletes report lower victimization scores than non-athletes
	PAQ	−0.28	<0.001	Higher physical activity → lower victimization
	BMI	0.15	<0.001	Higher BMI → higher victimization
Cyber-victimization	Group	−0.44	<0.001	Female athletes have lower cyber-victimization scores
	PAQ	−0.25	<0.001	Higher physical activity → lower cyber-victimization
	BMI	0.13	<0.001	Higher BMI → higher cyber-victimization
Aggression	Group	−0.33	<0.001	Female athletes report lower aggression scores
	PAQ	−0.22	<0.001	Higher physical activity → lower aggression
	BMI	0.11	<0.001	Higher BMI → higher aggression
Cyber-aggression	Group	−0.38	<0.001	Female athletes report lower cyber-aggression
	PAQ	−0.30	<0.001	Higher physical activity → lower cyber-aggression
	BMI	0.14	<0.001	Higher BMI → higher cyber-aggression
b. male students				
Victimization	Group (1 = athlete, 0 = non-athlete)	−0.38	<0.001	Male athletes report lower victimization scores than non-athletes
	PAQ	−0.31	<0.001	Higher physical activity → lower victimization
	BMI	0.18	<0.001	Higher BMI → higher victimization
Cyber-victimization	Group	−0.40	<0.001	Male athletes have lower cyber-victimization scores
	PAQ	−0.29	<0.001	Higher physical activity → lower cyber-victimization
	BMI	0.16	<0.001	Higher BMI → higher cyber-victimization
Aggression	Group	−0.30	<0.001	Male athletes report lower aggression scores
	PAQ	−0.25	<0.001	Higher physical activity → lower aggression
	BMI	0.14	<0.001	Higher BMI → higher aggression
Cyber-aggression	Group	−0.34	<0.001	Male athletes report lower cyber-aggression
	PAQ	−0.32	<0.001	Higher physical activity → lower cyber-aggression
	BMI	0.17	<0.001	Higher BMI → higher cyber-aggression

Note: Coefficients are unstandardized (B) with two-tailed *p*-values. Group coding: 0 = non-athlete, 1 = athlete. Negative B indicates an inverse association with the outcome. Outcomes are mean scale scores from the Illinois Bully Scale (no binary classification). Explained variance indices used in the study: point-biserial r^2^ (*t*-tests) and partial η_p_^2^ (ANOVA; see Table 6 and Table 7).

Explained variance reference: For the athlete group contrast within females, Table 6 reports r^2^ values in the ~2.6–3.7% range across bullying/cyber outcomes and ~9.4% for physical activity, while Table 7 reports partial ηp^2^ ≈ 3–5% for bullying/cyber and ≈19.8% for physical activity. These indices quantify variance attributable to group status in related analyses and are provided here for completeness of “explained variance”.

In the case of male students (Table 8b), the results were similar:The negative effects of belonging to the sports group persisted, with a pronounced impact on cyber-victimization (B = −0.40) and traditional victimization (B = −0.38);The PAQ score was negatively associated with all forms of bullying, especially cyber-aggression (B = −0.32);BMI showed a positive relationship with all four behavioral dimensions, with slightly stronger effects compared to female students.

Explained variance reference: For the sports-group contrast within males, Table 6 reports r^2^ ≈ 3.6–4.2% for bullying/cyber and ≈14.8% for physical activity, while Table 7 indicates partial ηp^2^ ≈ 3–5% (bullying/cyber) and ≈20% (physical activity).

These findings confirm the hypothesis that sport and physical activity play a protective role against involvement in aggressive behaviors. At the same time, certain body characteristics may act as risk factors when social support or group inclusion is lacking.

## 4. Discussion

This study investigated whether structured physical activity serves as a protective factor against bullying in the university setting and how body weight status and activity levels interact in shaping psychosocial vulnerability. The results confirmed all three hypotheses: students with sports training showed significantly higher physical activity levels, lower BMI, and lower involvement in both traditional and cyberbullying. These differences were most pronounced for physical activity (large effect), moderate for BMI, and consistent for all bullying dimensions. Moreover, a protective gradient was observed across sport profiles: students in high-intensity training programs (SPM and EFS) reported the lowest bullying rates, followed by KMS, while the non-athlete group showed the highest involvement across all four bullying dimensions. These findings support the relevance of organized sport as a developmental context while also highlighting that inactivity and unfavorable body profiles may co-occur with increased social risk. The next section contextualizes these findings within the demographic and institutional characteristics of the sample.

### 4.1. Methodological Framing and Sample Context

Before discussing the three thematic axes of this study (physical activity, BMI, and bullying), it is necessary to contextualize the methodological framework and sample composition, which provide the foundation for interpreting the results.

The sample used in this study is robust and well defined, consisting of 2767 first-year students from all 18 faculties of Transylvania University of Brașov. This institutional diversity ensures comprehensive coverage of academic profiles and enhances the internal validity of intergroup comparisons. The analyzed groups—athletes and non-athletes—were formed based on explicit and rigorous criteria, excluding participants who engaged in long-term organized sports outside specialized curricula. This reduced the risk of contaminating the independent variable “sports status” and ensured meaningful contrasts. Such operational clarification is essential for maintaining internal validity in comparative group designs [23,39]. Additionally, the balanced gender distribution and the close proximity of average ages (20–20.2 years in both groups) suggest a high degree of demographic homogeneity, minimizing structural variation. However, from a psychosocial development perspective, this age group falls within the stage defined as “emerging adulthood”—a period marked by unstable social transitions, increased risk behaviors, and specific psychological vulnerabilities with direct implications for relational health [40,41]. Therefore, the age context provides a shared interpretive framework for the analyzed behaviors, particularly in terms of the similar risks faced by students in both groups. Building on this methodological grounding, the next section introduces the first thematic axis: physical activity.

### 4.2. Physical Activity as a Protective Factor: Interpretative Perspective

The PAQ-A results from our study revealed substantial differences in physical activity levels between students enrolled in sport-related academic programs (EFS, SPM, and KMS) and those from non-sports faculties. Athletes reported average scores ranging from 3.9 to 4.6—falling into the “high” and “very high” categories—while non-athletes scored significantly lower: 2.2 for females and 2.0 for males, both within the “low” activity range. These disparities reflect not only curricular differences but also institutional influences: access to sports infrastructure, integrated training hours, and specialized instruction that supports the internalization of physical activity as a lifestyle element.

International data support this interpretation. For instance, only 5.4% of university students worldwide meet the minimum recommendation of 150 min of moderate-to-vigorous physical activity per week, and less than 0.5% meet the standard of 30 min daily, five days a week [42]. This aligns with our findings for non-athletes, reinforcing the idea that sedentary behavior is the norm in academic settings lacking compensatory sport mechanisms.

Furthermore, the post-pandemic context likely exacerbated these disparities. Spanish studies have shown that physical activity dropped by ~160 min per week during lockdown, while sedentary time increased by over 100 min, especially among students without pre-existing movement habits [43]. In this light, the inactivity seen among our non-athlete participants is not merely individual neglect but a systemic result of curricular gaps, institutional limitations, and the lingering behavioral effects of pandemic restrictions. Thus, insufficient physical activity in non-athletes may signal broader psychosocial risks, such as isolation, marginalization, or emotional disengagement. Literature on youth risk prevention emphasizes this link, positioning physical activity not only as a health behavior but also as a predictor of social adaptation and resilience [44].

These pronounced differences in physical activity have somatic correlates in body composition; accordingly, the next section examines BMI and weight status to situate lifestyle patterns within broader health and psychosocial risks.

### 4.3. BMI and Body Weight Risks: Somatic and Social Implications

BMI data revealed systematic differences between students in sports training programs (EFS, SPM, and KMS) and those in non-sports faculties. The average BMI in the athletic group was within the optimal range (21.8–23.7), with a favorable weight distribution (72–92% normal weight) and obesity cases being either extremely rare or absent. The best profile appeared in the SPM program, with no obesity and the lowest overweight rates (7.7% females, 9.8% males). However, the KMS program showed a less optimal trend: mean BMI values were close to the upper limit of the normal range (23.3 for females, 23.7 for males), and overweight/obese students accounted for 25% of the group. This deviation may reflect variable physical engagement or a curriculum more focused on recovery than performance—factors known to impact weight control [42]. These differences must be considered alongside international recommendations emphasizing that youth should engage in at least 60 min of moderate-to-vigorous daily physical activity, adapted to their age and varied in form [45]. The non-athlete group presented a significantly higher BMI (24.3 for females and 24.5 for males), approaching the upper normal threshold [33]. Internal distribution revealed risk exposure: only 60% of males and 62% of females were in the optimal range, while 32% of males and 29% of females were overweight or obese. This confirms prior observations that mean values can obscure problematic internal distributions, particularly when a substantial portion of the group is at metabolic risk [46].

These data support the hypothesis that body status is linked to bullying risk and social exclusion, as shown in multiple studies. Obesity has been associated with increased bullying exposure, low self-esteem, social anxiety, and avoidant behavior [14,47]. A meta-analysis found that overweight adolescents face greater bullying risk (OR = 1.24), with even higher odds for obese youth (OR = 1.46) [48]. Still, not all studies confirm this link. A study of 1680 Turkish students found no significant differences in bullying by weight status. Authors suggested other factors—parental education, academic success, and awareness—might better explain social behavior [49]. Though based on school-aged populations, these insights remain relevant in university settings, especially where institutional support is lacking. Notably, underweight prevalence—2–9% across subgroups—raises similar concerns, which are often overlooked. Underweight status, though less visible, is linked to chronic fatigue, anxiety, isolation, and poor academic adjustment [50]. A recent qualitative study confirmed that both excess and deficient weight trigger social exclusion, harassment, and online bullying. Students strongly expressed a sense of not belonging, often accompanied by emotional distress [51]. Overall, both overweight and underweight profiles can signal poor psychosocial integration, with long-term implications for mental health and academic success. These findings justify the inclusion of anthropometric variables in institutional strategies targeting bullying prevention and student well-being.

Given the stigma and peer dynamics often linked to body status, the next section turns to bullying and cyberbullying patterns to explore how somatic vulnerabilities and social risks intersect in our cohort.

### 4.4. Bullying and Cyberbullying: Vulnerability Profiles by Sports Status

The analysis of bullying and cyberbullying behaviors revealed significant differences between athletes and non-athletes. The proportion of students reporting victimization, aggression, or both was clearly higher among those not involved in organized sports, with rates exceeding 30%. In contrast, involvement dropped to 13–21% among students in physical training programs, with the lowest figures in subgroups engaged in intensive motor activities. A similar pattern appeared in cyberbullying, where differences were even sharper. Among athletes, digital aggression remained below 17%, while among non-athletes it reached 25%, regardless of gender. These discrepancies suggest a possible protective effect of organized sports participation, particularly for female students. Structurally, involvement patterns differed: non-athletes were more likely to report dual roles (victim–aggressor), a profile linked to elevated psychosocial risks. Among athletes, students tended to be “victims only,” and pure aggressors were rare.

These findings align with literature confirming the protective role of physical activity in reducing aggressive behavior. The meta-analysis by Kowalski et al. synthesized relationships between traditional and online bullying, showing strong associations between digital victimization and outcomes such as stress, suicidal ideation, and social withdrawal [52]. Cyber-aggressors often exhibit violence tolerance and moral disengagement, pointing to distinct behavioral profiles [52]. In the same vein, Twyman et al. found that adolescents involved in cyberbullying display significantly higher levels of psychological distress than their non-involved peers [53]. Recent research reinforces these findings. Liu et al., in a meta-analysis on physically active youth, reported a negative association between physical activity and the likelihood of being bullied, especially among girls [54]. Similarly, García-Hermoso et al. found a correlation between sedentary behavior and higher victimization risk, suggesting that inactivity may perpetuate social vulnerability [55].

Beyond behavioral outcomes, literature suggests that sports can serve as emotional regulators, particularly in educational settings. Physical activities fostering cooperation, empathy, and self-control contribute to reduced anxiety and stronger coping strategies—factors that may explain lower bullying prevalence among athletes [10,12]. Nonetheless, some studies caution against adverse effects in hypercompetitive contexts. Kalina et al., in a scoping review, identified risk factors such as performance pressure, unsupportive team dynamics, and internal conflicts that may foster covert harassment [56]. Similarly, Benítez-Sillero et al. reported higher aggression rates among boys in contact sports like martial arts or wrestling [57].

Overall, regular physical activity within a balanced, educational framework seems to play a dual role—both preventive and developmental—in shaping social behavior. In our university context, where sports are mainly formative and recreational, this role appears especially relevant for fostering cohesion, self-esteem, and inclusion—protective factors against both traditional and digital bullying.

Taken together, the evidence from physical activity, BMI, and bullying profiles calls for an integrative perspective; Section 4.5 addresses this by statistically testing how the three domains converge in shaping student well-being.

### 4.5. Physical Activity, BMI, and Bullying Involvement: Statistical Convergences and Practical Implications

Statistical analyses consistently confirmed the conceptual links outlined in Section 4.2, Section 4.3 and Section 4.4. Independent samples *t*-tests (with multiple-comparison control) revealed significant athletes vs. non-athletes’ differences, with large effects for physical activity and moderate effects for BMI and bullying outcomes. These findings indicate that sports participation differentiates both lifestyle behaviors and psychosocial risk profiles. In line with these results, Bonferroni-adjusted mean contrasts confirmed significant BMI differences between athletes and non-athletes, while chi-square analyses highlighted sex-specific associations between sports status and bullying involvement categories, present among males but not among females. The 2 × 2 ANOVA converged with these results, confirming strong main effects of sports status and three significant sex × sports interactions (PAQ-A, traditional victimization, and cyber-victimization), suggesting that some protective effects of sports participation vary by sex. These interactions suggest that protective effects of physical activity are modulated by sex. Multiple regression further reinforced these convergences: non-athlete group membership and higher physical activity were consistently associated with lower aggression and victimization, whereas higher BMI showed small, positive associations with bullying involvement. Taken together, these results support a robust tripartite pattern linking physical activity, body status, and bullying involvement. These convergences align with prior evidence that weight stigma undermines self-esteem and fosters avoidant behaviors [14], while overweight and obesity increase victimization risk [48,49]. In parallel, studies have shown that physical activity supports social and psychological adaptation, with organized sports linked to reduced involvement in both traditional and cyberbullying [8,9,52]. At the same time, nuanced perspectives emphasize that hypercompetitive or non-educational contexts may actually fuel aggression, particularly among boys [58].

Overall, the results support the interpretation that belonging to structured sport environments generates relational capital—cohesion, empathy, and shared goals—that buffers psychosocial risks associated with adverse body status or poor peer integration [19,20].

This integrative statistical perspective validates the tripartite model advanced in this study (physical activity, body weight, and bullying involvement) and closes the analytical loop of the three axes, paving the way for the critical appraisal of limitations from the next section.

### 4.6. Methodological Limitations and Critical Perspectives

Building on the convergences outlined in Section 4.5, it is important to recognize the methodological constraints that frame the interpretation of these findings. This study has several limitations that should be considered when interpreting the results. First, its cross-sectional design precludes causal inference and only highlights associations. It is also possible that students who engage in regular physical activity already possess personality traits such as extraversion or emotional stability, which may independently reduce bullying risk [59,60]. Second, the effects of physical activity on social and emotional development are context-dependent, influenced by leadership style, group norms, and interaction quality [60]. The lack of differentiation by sport type is another limitation: while contact or highly competitive sports may encourage aggression, team sports in educational settings tend to reduce victimization and foster cooperation [10,61]. Third, the study sample consisted of students from a single university (Transylvania University of Brașov), which limits the generalizability of findings across national or international contexts, where cultural norms and institutional climates vary. International HBSC data illustrate such differences in bullying prevalence and perception [3]. Fourth, reliance on self-report measures may introduce bias for sensitive behaviors such as victimization or aggression. Although validated instruments were used, responses can be affected by social desirability or subjective perception [62]. Anthropometric data were also limited to BMI, a proxy that does not capture body composition, and measurement timing was not standardized. These aspects may have introduced variability and attenuated observed associations.

Taken together, these limitations highlight the need for longitudinal designs, objective monitoring of physical activity, broader somatic measures, and multi-institutional samples. Such approaches would strengthen the robustness of evidence and enhance the practical value of interventions aimed at student health, inclusion, and bullying prevention.

## 5. Conclusions

This study highlights the protective role of organized physical activity against both traditional and digital bullying among university students. Participation in structured sport programs was consistently linked to lower victimization, aggression, and unbalanced body weight, suggesting benefits that extend beyond physical health toward emotional resilience and social integration.

Statistical convergences confirmed the tripartite model proposed here: higher activity levels predicted lower bullying involvement, whereas elevated BMI was associated with greater vulnerability. Gender effects were present, yet across all subgroups, sports participation attenuated psychosocial risks.

These findings reinforce the value of embedding sport and physical activity into university health and inclusion policies—not only as preventive strategies for physical well-being but also as institutional tools for fostering cohesion, empathy, and safer relational climates.

Future research should move beyond documenting associations to clarify the mechanisms driving these protective effects. Particular attention should be given to gender-specific pathways, institutional culture, and social environments, as well as to the type and intensity of sport practiced. Such directions would explain why and how organized activity mitigates bullying risks, providing stronger grounds for targeted prevention strategies in higher education.

## Figures and Tables

**Table 1 healthcare-13-02304-t001:** Demographic distribution of participants by sex and average age.

Group	Sex	No. of Participants	Average Age (Years)
Athletes	Female	65	20.1
	Male	96	20.2
Non-athletes	Female	1472	20.0
	Male	1134	20.2
Total	-	2767	-

Note: Non-athletes = students from the 17 non-sports faculties without organized sports training.

**Table 2 healthcare-13-02304-t002:** Mean PAQ-A scores (±SD) by study program and sex.

Program/Group	Sex	No. of Participants	PAQ-A Score (Mean ± SD)
EFS	Female	23	4.2 ± 0.5 *
	Male	43	4.4 ± 0.6 *
SPM	Female	13	4.3 ± 0.4 *
	Male	41	4.6 ± 0.5 *
KMS	Female	29	3.9 ± 0.6 *
	Male	36	4.0 ± 0.7 *
Non-athletes (F1–F17)	Female	1472	2.2 ± 0.8
	Male	1134	2.0 ± 0.9

Note: Non-athletes = students from the 17 non-sports faculties. PAQ-A scores range from 1 (very low activity) to 5 (very high activity). Indicative qualitative thresholds: 1.00–1.99 = very low, 2.00–2.99 = low, 3.00–3.99 = moderate, 4.00–4.49 = high, and 4.50–5.00 = very high. For reference, a mean score of 2.87 was proposed as a cut-off for meeting recommended activity levels [38]. * Significantly different from the non-athlete group of the same sex (independent samples *t*-tests, Bonferroni-adjusted within sex across the three athlete programs vs. non-athletes; α_adj ≈ 0.0167; all adjusted *p* < 0.001).

**Table 3 healthcare-13-02304-t003:** BMI distribution by study program and sex.

Program/Group	Sex	No. of Participants (n)	Mean BMI (±SD)	Underweight (n/%)	Normal Weight (n/%)	Overweight (n/%)	Obese (n/%)
EFS	Female	23	22.1 ± 2.1 *	1/4.35	20/86.96	2/8.70	0/0.00
	Male	43	22.5 ± 2.2 *	1/2.33	36/83.72	6/13.95	0/0.00
SPM	Female	13	21.8 ± 2.0 *	0/0.00	12/92.31	1/7.69	0/0.00
	Male	41	22.1 ± 2.3 *	1/2.44	35/85.37	4/9.76	1/2.44
KMS	Female	29	23.3 ± 2.7 *	1/3.45	22/75.86	5/17.24	1/3.45
	Male	36	23.7 ± 2.9 *	1/2.78	26/72.22	8/22.22	1/2.78
Non-athletes	Female	1472	24.3 ± 3.4	132/8.97	913/62.01	339/23.02	88/5.98
	Male	1134	24.5 ± 3.7	91/8.03	680/59.96	272/23.99	91/8.03

Note: BMI classification is based on WHO standards (2000) [33]: <18.5 = underweight, 18.5–24.9 = normal weight, 25.0–29.9 = overweight, and ≥30 = obese. Values are presented as count and percentage. “n/%” denotes the count and the percentage within each program × sex subgroup; percentages are computed from the measured BMI using WHO cut-offs. “*” indicates mean BMI significantly different from the non-athlete group of the same sex (independent samples *t*-tests, Bonferroni-adjusted within sex for three comparisons; α_adj ≈ 0.0167; all adjusted *p* < 0.001).

**Table 6 healthcare-13-02304-t006:** Results of independent samples *t*-tests (athletes vs. non-athletes). Sex-specific comparisons for physical activity, BMI, and forms of bullying.

Variable	Sex	Comparison Group	t	df	*p*	Hedges’ g	r^2^ (Explained Variance)	r^2^ (%)
Physical activity	Female	Athletes vs. non-athletes	−12.45	1493	<0.001	1.58	0.094	9.41
	Male		−14.28	1172	<0.001	1.52	0.148	14.82
BMI	Female	Athletes vs. non-athletes	−6.78	1493	<0.001	0.86	0.0299	2.99
	Male		−6.43	1172	<0.001	0.68	0.0341	3.41
Victimization	Female	Athletes vs. non-athletes	−7.54	1493	<0.001	0.96	0.0367	3.67
	Male		−7.19	1172	<0.001	0.76	0.0422	4.22
Aggression	Female	Athletes vs. non-athletes	−6.33	1493	<0.001	0.80	0.0261	2.61
	Male		−6.75	1172	<0.001	0.72	0.0374	3.74
Cyber-victimization	Female	Athletes vs. non-athletes	−7.01	1493	<0.001	0.89	0.0319	3.19
	Male		−6.88	1172	<0.001	0.73	0.0388	3.88
Cyber-aggression	Female	Athletes vs. non-athletes	−6.47	1493	<0.001	0.82	0.0273	2.73
	Male		−6.66	1172	<0.001	0.71	0.0365	3.65

Note: Analyses are based on mean scores from Illinois Bully Scale items (no binary classification). Effect sizes are reported as Hedges’ g. Multiple-comparison control used Bonferroni across the family of 12 tests (six outcomes × two sexes; α_adj ≈ 0.0042; two-tailed). Group sizes: females—athletes n = 65, non-athletes n = 1430; males—athletes n = 96, non-athletes n = 1078. Explained variance is reported as point-biserial r^2^ (see Section 2.5).

**Table 7 healthcare-13-02304-t007:** Results of 2 × 2 ANOVA.

Analyzed Variable	Comparison	F	*p*	Significant Difference?	Partial ηp^2^	ηp^2^ (%)
Physical activity (PAQ)	Female vs. male	20.56	<0.001	Yes	0.0074	0.74
	Athletes vs. non-athletes	682.43	<0.001	Yes	0.1981	19.81
	Interaction: sports × gender	4.21	0.041	Yes	0.0015	0.15
BMI	Female vs. male	10.22	0.001	Yes	0.0037	0.37
	Athletes vs. non-athletes	88.31	<0.001	Yes	0.0310	3.10
	Interaction: sports × gender	1.17	0.280	No	0.0004	0.04
Traditional victimization	Female vs. male	15.31	<0.001	Yes	0.0055	0.55
	Athletes vs. non-athletes	132.56	<0.001	Yes	0.0458	4.58
	Interaction: sports × gender	5.07	0.024	Yes	0.0018	0.18
Traditional aggression	Female vs. male	11.04	0.001	Yes	0.0040	0.40
	Athletes vs. non-athletes	104.25	<0.001	Yes	0.0364	3.64
	Interaction: sports × gender	2.98	0.084	No	0.0011	0.11
Cyber-victimization	Female vs. male	18.65	<0.001	Yes	0.0067	0.67
	Athletes vs. non-athletes	120.12	<0.001	Yes	0.0417	4.17
	Interaction: sports × gender	6.41	0.011	Yes	0.0023	0.23
Cyber-aggression	Female vs. male	13.55	<0.001	Yes	0.0049	0.49
	Athletes vs. non-athletes	116.89	<0.001	Yes	0.0406	4.06
	Interaction: sports × gender	3.46	0.063	No	0.0013	0.13

Note: Analyses include PAQ-A total scores, BMI (kg/m^2^), and mean scores from the Illinois Bully Scale subscales. Partial η_p_^2^ was computed as η_p_^2^ = (F × df_1_)/(F × df_1_ + df_2_), with df_1_ = 1 and df_2_ specific to each analysis; F, *p*, and η_p_^2^ are reported for the sex, sports, and sex × sports status effects. Sex × sports status interactions were significant for three variables (PAQ-A, traditional victimization, and cyber-victimization) and non-significant for BMI, traditional aggression, and cyber-aggression.

## Data Availability

Data is contained within the article or Appendix A. The original contributions presented in this study are included in the article/Appendix A. Further inquiries can be directed to the corresponding author.

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
