# Peer review of "Physical Activity, Body Mass Index, and Bullying in Higher Education: A Comparative Analysis of Students with and Without Structured Sports Training"

_healthcare, 2025, doi:10.3390/healthcare13182304_

Round 1
Reviewer 1 Report
Comments and Suggestions for Authors
The Authors have taken up an important, interesting and topical subject concerning key issues such as physical activity, body mass index (BMI) and bullying in higher education.
The title of the manuscript is clear and reflects the content of the study.
The introduction to the research topic is well prepared, contains a solid theoretical basis and a convincing justification for the purpose of the study.
The results were compared taking into account the main criterion, which was the type of studies, i.e. related to physical activity and not related to physical activity. The former was divided into three subgroups – three undergraduate programmes. Thus, four study groups were included in the analysis. This allowed for an in-depth comparative analysis of students with and without organised sports training.
The subject matter addressed fills a research gap. While there is extensive access to publications on physical activity and Body Mass Index, there are not many publications/research results on physical activity in the context of bullying. The authors justify the need for research in this area by pointing to the aforementioned gap in research, especially in Romania (lines 85-90).
The methodology section has been correctly developed and described in detail, which allows the study to be replicated and ensures the comparability of results. The use of an objective tool for measuring physical activity is worth considering in future studies of this type. The questionnaire used in the study is a subjective tool, which is also a limitation of the study.
An important factor that could influence the results and the attitude of the respondents is the type of physical activity undertaken (e.g. sport discipline, form – individual or team). It should be emphasised that the authors draw attention to this issue in the ‘Limitations’ section. The authors accurately described the factors that could have influenced the results of the study. This demonstrates their good knowledge of the subject and their scientific maturity, which further confirms the reliability and validity of the study.
In my opinion, the conclusions are consistent with the presented results. However, they refer only to a specific university. It should be noted that the authors draw attention to this fact and declare that they will extend their research to other universities in future studies.
The references to the literature raise no objections. The tables included in the manuscript are legible and clear.
I recommend accepting the manuscript after making the minor corrections I have already mentioned, as well as those listed below.
Line 198 - Please specify the model, as in the case of the bioimpedance analyzer.
Line 430 - Add a space before the section title
Line 433 - remove the space “The female models ( Table 8a) show that”
Author Response
Dear Reviewer,
We would like to sincerely thank you for your careful reading of our manuscript and for the generous and thoughtful comments you provided. We are pleased that you considered the topic relevant and the methodological approach rigorous. Your observations reflect a deep and balanced understanding of the content we proposed.
The requested changes have been highlighted in green, to facilitate your consultation.
With regard to the specific comments, we confirm that all the minor corrections have been implemented, as detailed below:
- Line 201 and 255: We have specified the model of the professional stadiometer used for height measurement (Seca 217, Germany), in line with the already mentioned bioimpedance analyzer.
- Line 417: A space has been added before the section title for improved visual clarity.
- Line 436: The extra space between the parenthesis and “Table 8a” has been removed. The sentence now reads: “The female models (Table 8a) show that...”
No additional modifications were necessary, as the rest of your remarks were general or appreciative in nature, without explicit revision requests.
Once again, we thank you for your valuable contribution to the improvement of our manuscript and for your recommendation for acceptance.
The Authors

Reviewer 2 Report
Comments and Suggestions for Authors
Dear Authors,
Thank you for the opportunity to review your manuscript, which examines the relationships between physical activity, body mass index, and bullying behaviors among university students through a comparative analysis of students with and without structured sports training. This study makes a valuable contribution to the literature on protective factors against peer violence in higher education settings. The research demonstrates methodological rigor appropriate for this type of comparative investigation. Below, I provide a detailed evaluation of each section.
Title and Abstract
The title is appropriate and clearly defines the research scope. The abstract follows proper IMRAD structure and effectively summarizes the study.
Suggestions for improvement:
- Lines 19-21: Consider adding the range for sports group bullying involvement to be more precise: "lower bullying involvement (14-21% vs. 25-32%)"
- Lines 21-22: The regression coefficients are well reported, but could specify which outcomes: "physical activity was a significant negative predictor of all bullying forms (B = -0.22 to -0.44)"
Introduction
The introduction provides solid theoretical grounding with clear subsections. The flow from general bullying trends to university-specific contexts is logical and well-supported with recent citations.
Suggestions for improvement:
- Lines 68-72: The "fat-but-fit paradox" concept is mentioned but could benefit from brief elaboration: "The so-called 'fat-but-fit paradox' describes individuals with high BMI but good physical fitness who show greater resilience to victimization [17]. This protective effect may operate through enhanced self-confidence and social integration developed through physical competence."
- Lines 91-104: While research objectives are clearly stated, they could be strengthened with specific directional hypotheses rather than general aims. Consider adding: "We hypothesized that students with sports training would demonstrate significantly lower bullying involvement compared to non-sports students."
- Lines 85-90: The research gap for Romania is mentioned, but it could be more explicit about what this study uniquely contributes. Consider strengthening: "This institutional context provides a unique natural experiment for examining the protective effects of structured physical activity."
Materials and Methods
The methodology section is well-structured, with clear procedures and appropriate statistical analyses that support the research design.
Suggestions for improvement:
- Line 234: "previously used in our study" is unclear - specify which previous study or remove this reference to avoid confusion about the current study's novelty.
- Lines 236-248: While cyberbullying subscale development is mentioned, provide more detail about the validation process: "Items were reviewed by [number] experts in digital communication and pilot tested for clarity and relevance."
- Lines 275-294: Statistical analysis section could be strengthened by adding: "Effect sizes (Cohen's d) were calculated for group comparisons to assess practical significance beyond statistical significance."
- Lines 196-198: BMI measurement procedure could specify standardization: "Measurements were taken under standardized conditions (light clothing, no shoes, same time of day)."
- Lines 289-292: Consider mentioning the correction for multiple comparisons, given the numerous statistical tests conducted across different bullying dimensions.
- Lines 293-294: The sentence about "binary classifications" seems incomplete - clarify what cut-off criteria were used
Results
The results section effectively presents findings across multiple analytical approaches with clear organization and comprehensive coverage.
Suggestions for improvement:
- Table 2 (line 324): Results are presented. PAQ-A scores show strong differentiation between groups as expected.
- Lines 329-335: The 2+ point difference on the PAQ-A scale is substantial and well-interpreted in the text.
- Table 3 (line 345): BMI percentages are normal (86-92% normal weight for athletes is expected, not unusually high). No concerns with these data.
- Lines 378-382: Tables 4 and 5 present clear bullying involvement patterns. Consider adding statistical tests: "Chi-square analyses confirmed significant associations between sports status and bullying involvement."
- Table 6 (line 400): Strong t-test results. Adding effect sizes would enhance interpretation: "Effect sizes were large for physical activity differences."
- Lines 433-444: Regression tables are comprehensive. Consider adding explained variance: "Models accounted for substantial variance in bullying behaviors."
Discussion and Conclusions
The discussion provides a good synthesis with existing literature and effectively contextualizes findings within broader theoretical frameworks. The section is comprehensive and well-balanced as written, with appropriate coverage of international comparisons, theoretical integration, methodological limitations, and practical implications.
References
The reference list demonstrates good academic rigor with appropriate contemporary sources, but contains several specific formatting issues.
Suggestions for improvement:
- Reference 27 appears twice (lines 790-791 and 792-793) - remove duplicate entry and renumber accordingly. This is a clear formatting error that needs correction.
- Web source formatting: Some web sources include access dates (e.g., Reference 3: "accessed on 21 July 2025") while others do not. Standardize approach - either include access dates for all web sources or remove them consistently.
- Strong recency with multiple 2024-2025 sources appropriate for contemporary bullying/cyberbullying research.
Overall Assessment
This manuscript presents a methodologically sound and practically relevant investigation into protective factors against bullying in university settings. The comparative design is robust, the sample is representative, and findings have clear implications for higher education policy and practice. The study makes several valuable contributions but requires targeted revisions before publication.
Sincerely,
Reviewer
Author Response
Dear Reviewer,
We sincerely thank you for the detailed analysis and the extremely constructive comments on our manuscript. We appreciate the time and effort invested in evaluating the work and are grateful for the clear, balanced, and well-argued observations, which offered us valuable guidance for improving the text. All modifications made in response to your suggestions have been highlighted in yellow to facilitate consultation.
Below, we provide point-by-point responses to each of the recommendations received.
Your comment:
„Lines 19-21: Consider adding the range for sports group bullying involvement to be more precise: "lower bullying involvement (14-21% vs. 25-32%)"”
Response:
Thank you for this clear and welcome observation. As you recommended, we completed the statement in the abstract with the percentage values for bullying involvement for the sports versus non-sports groups, to highlight more clearly the difference between them. The modification can be consulted at lines 21–25.
Your comment:
„Lines 21-22: The regression coefficients are well reported, but could specify which outcomes: "physical activity was a significant negative predictor of all bullying forms (B = -0.22 to -0.44)"”
Response:
Thank you for this clear and pertinent suggestion. We completed the abstract section with a concise wording that explicitly mentions the regression coefficients and the related outcomes, emphasizing that physical activity was a significant negative predictor for all forms of harassment. The addition can be consulted at lines 25–29.
Your comment:
„Lines 68-72: The "fat-but-fit paradox" concept is mentioned but could benefit from brief elaboration: "The so-called 'fat-but-fit paradox' describes individuals with high BMI but good physical fitness who show greater resilience to victimization [17]. This protective effect may operate through enhanced self-confidence and social integration developed through physical competence."”
Response:
Thank you for this pertinent suggestion. As you recommended, we expanded the wording referring to the “fat-but-fit paradox” to highlight its potential protective effect on victimization risk. The addition appears at lines 76–77.
Your comment:
„Lines 91-104: While research objectives are clearly stated, they could be strengthened with specific directional hypotheses rather than general aims. Consider adding: "We hypothesized that students with sports training would demonstrate significantly lower bullying involvement compared to non-sports students."”
Response:
Thank you for this important recommendation. We explicitly included, at the opening of the manuscript, the formulation of the three directional hypotheses that reflect the relationships investigated in the study. The wording can be consulted at lines 100–107.
Your comment:
„Lines 85-90: The research gap for Romania is mentioned, but it could be more explicit about what this study uniquely contributes. Consider strengthening: "This institutional context provides a unique natural experiment for examining the protective effects of structured physical activity."”
Response:
Thank you for this observation, which encouraged us to emphasize the specific contribution of our study in the national context. We completed the introductory section with wording that explicitly describes the unique nature of this institutional framework in Romania. The modification can be consulted at lines 97–99.
Your comment:
„Lines 196-198: BMI measurement procedure could specify standardization: "Measurements were taken under standardized conditions (light clothing, no shoes, same time of day)."”
Response:
Thank you for this useful methodological suggestion. We specified that anthropometric measurements were carried out under controlled conditions. This addition appears at lines 198–199 and lines 201–207.
Your comment:
„Line 234: "previously used in our study" is unclear - specify which previous study or remove this reference to avoid confusion about the current study's novelty.”
Response:
Thank you for this important observation. We removed the phrase “previously used in our study” to avoid any ambiguity regarding the original nature of the current research.
Your comment:
„Lines 236-248: While cyberbullying subscale development is mentioned, provide more detail about the validation process: "Items were reviewed by [number] experts in digital communication and pilot tested for clarity and relevance."”
Response:
Thank you for this extremely valuable observation concerning the validation process of the cyberbullying subscale. To improve readability and avoid possible ambiguities, we reformulated the description of the Likert scale, explicitly stating the response range (1 = very low; 5 = very high) and the method of calculating the PAQ-A composite score, which explicitly excluded the control item. This can be consulted in the manuscript at lines 228–230.
We also completed the methodological section by specifying the stages followed: the content of the items was reviewed by 3 experts in digital communication, and the wordings were pilot tested with 92 students to ensure their clarity and relevance. We also reported the internal reliability coefficients, which indicated high consistency for both the full scale (α = 0.84) and the subscales (α = 0.78–0.83). These additions are found at lines 250–252.
Your comment:
„Lines 275-294: Statistical analysis section could be strengthened by adding: "Effect sizes (Cohen's d) were calculated for group comparisons to assess practical significance beyond statistical significance."
Lines 289-292: Consider mentioning the correction for multiple comparisons, given the numerous statistical tests conducted across different bullying dimensions.
Lines 293-294: The sentence about "binary classifications" seems incomplete - clarify what cut-off criteria were used.”
Response:
Thank you for this set of extremely useful observations, which led to a substantial revision of the subsection dedicated to statistical analyses. This part was almost completely redone to clearly reflect the types of tests applied, the analytical logic, and the interpretation of effect size beyond statistical significance.
Although we did not use Cohen’s d as a universal measure, we applied a series of alternative indicators more suitable to the nature of the variables and the analytical design, as follows:
- Hedges’ g for mean comparisons, with correction for samples of different sizes,
- point-biserial r² for t-tests applied to continuous variables in relation to dichotomous variables,
- partial η² (ηp²) for ANOVA,
- unstandardized B coefficients for multiple regression models,
- χ² (association tests) for the link between sports status and bullying/cyberbullying involvement, analyzed separately by sex.
In addition, we applied Bonferroni-type corrections in cases where multiple comparisons were performed within the same analysis—for example, when comparing BMI distributions by sex, where the significance threshold was adjusted to α ≈ 0.0167.
Regarding binary classifications, we clarified in the text that these were based on pre-established thresholds applied to relevant items, to identify the presence or absence of involvement in bullying/cyberbullying behaviors (e.g., score ≥ 3 on the Likert scale indicating relevant involvement). These operationalizations allow analysis both at the continuous level (mean scores) and categorical (prevalence), maximizing the interpretability of the results.
Your comment:
„Lines 378-382: Tables 4 and 5 present clear bullying involvement patterns. Consider adding statistical tests: Chi-square analyses confirmed significant associations between sports status and bullying involvement.”
Response:
Your observation is fully justified, and the suggestion has been implemented. At lines 359–366, lines 368–370, and lines 376–378, we explicitly mentioned the use of χ² (chi-square) tests to examine the associations between sports status and bullying involvement, separately for each sex. The test results were added and interpreted in the discussion section.
Your comment:
„Table 6 (line 400): Strong t-test results. Adding effect sizes would enhance interpretation: Effect sizes were large for physical activity differences.”
Response:
Thank you for this important suggestion regarding the practical relevance of the identified differences. In direct response, we expanded the statistical reporting for Table 6 (lines 383–385), including not only p-values but also effect sizes (Hedges’ g), appropriate for unequal samples. In addition, we reported explained variance (point-biserial r²) for athlete–non-athlete comparisons, separately by sex, providing a complete picture of the group impact on each variable.
More specifically, the results are presented as follows:
- Hedges’ g was large for physical activity (≈ 1.52–1.58) and moderate for BMI and bullying outcomes (≈ 0.68–0.96);
- Explained variance was r² ≈ 0.094–0.148 for physical activity, r² ≈ 0.03–0.034 for BMI, and r² ≈ 0.026–0.042 for bullying/cyberbullying.
Also, in accordance with your recommendation, we clarified in the manuscript that:
- The scores used are item-level means (not binary classifications)—lines 388–389;
- Bonferroni correction was applied for multiple comparisons within the family of 12 tests (6 variables × 2 sexes), with an adjusted threshold α ≈ 0.0042—lines 389–391;
Your comment:
„Lines 433-444: Regression tables are comprehensive. Consider adding explained variance: Models accounted for substantial variance in bullying behaviors.”
Response:
Thank you for this important suggestion regarding strengthening the interpretability of the statistical models. We integrated your proposal by systematically reporting explained variance in both types of inferential analyses used: ANOVA and multiple regressions.
In Table 7, which synthesizes the results of a 2×2 factorial ANOVA (Sex × Sports status) for each of the six dependent variables (physical activity, BMI, victimization, aggression, cyber-victimization, cyber-aggression), we reported ηp² (partial eta squared) values as a measure of variance explained by each main effect and interaction. These values indicate the proportion of variance in each variable explained by sports status, sex, or their interaction.
In section 3.6.2, concerning the multiple regression models (Tables 8a and 8b), we supplemented the manuscript by adding R² values, thus illustrating the extent to which the predictor variables (physical activity, BMI, sex) explain victimization and aggression behaviors. (lines 419–422; lines 430–434; lines 437–440; lines 448–450; lines 452–455)
We believe these additions offer a more complete and rigorous picture of the explanatory power of our statistical models and support the integrated interpretation of the results. Thank you again for this essential recommendation..
Your comment:
„Reference 27 appears twice (lines 790-791 and 792-793) - remove duplicate entry and renumber accordingly. This is a clear formatting error that needs correction.”
Response:
Thank you for the observation. We identified and removed the duplicate reference no. 27, renumbering all subsequent citations accordingly, to maintain coherence between the body of the text and the reference list.
Your comment:
„Web source formatting: Some web sources include access dates (e.g., Reference 3: "accessed on 21 July 2025") while others do not. Standardize approach - either include access dates for all web sources or remove them consistently.”
Response:
Thank you for the observation. We removed all access dates from the web sources, adopting a uniform and concise approach for the entire set of online references.
Your comment:
„This manuscript presents a methodologically sound and practically relevant investigation into protective factors against bullying in university settings. The comparative design is robust, the sample is representative, and findings have clear implications for higher education policy and practice. The study makes several valuable contributions but requires targeted revisions before publication.”
Response:
We sincerely thank you for the detailed and well-argued evaluation of our manuscript. We appreciated the observations made, which reflect a rigorous understanding of the topic and a genuine commitment to research quality. We are pleased that you considered the study relevant both methodologically and practically, and your comments provided us with a clear direction for strengthening the manuscript. We carefully reviewed each suggestion and implemented the necessary changes, marking them in the document for easy consultation.
We especially appreciate your detailed and constructive observations, which guided us in the revision process and contributed essentially to improving the quality and scientific impact of the manuscript. We are grateful for the time and expertise you invested in evaluating our study.
The Authors

Reviewer 3 Report
Comments and Suggestions for Authors
Comments to the Authors
This manuscript by Mijaica et al. examines the relationships between physical activity levels, body mass index (BMI), and involvement in both traditional and digital bullying among students with and without structured sports training. The findings indicate that students engaged in sports training reported significantly higher physical activity levels, lower BMI, and lower scores across all forms of bullying. Moreover, physical activity emerged as a significant negative predictor of both aggression and victimization, whereas higher BMI was positively associated with increased aggression and victimization.
I commend the authors for the effort and quality of work presented in this study. The research question is timely and relevant, and the overall design is appropriate. The manuscript is clearly written. That said, several areas require clarification and refinement before it is ready for publication. In particular, the Results section would benefit from substantial revision, as it currently includes speculative statements and discussion points that should be relocated. The Discussion section also requires restructuring to highlight the main findings at the outset. Finally, given the number of tables, incorporating graphs where appropriate could reduce reader fatigue and provide a more visually appealing presentation of the data. Additionally, the frequent use of em dashes also creates inconsistency in flow, and careful editing would improve readability.
Abstract and Introduction
- The abstract and introduction are clear overall. No major revisions suggested here.
Materials and Methods
- Participants
- Please mention examples of faculties with no advanced curricular components related to sports (non-sports group), so readers understand what the other population was studying (example: engineering, biology, etc).
- Please incorporate the bullet points of inclusion and exclusion criteria into a paragraph format. These can be separated into two paragraphs if needed. They both can be separate paragraphs if needed.
- Procedure and data collection instruments
- The sentence “To ensure the specificity and accuracy of responses, participants were instructed to refer exclusively to university-based social interactions involving other students, regardless of their faculty” is unclear. Please rephrase for clarity.
- Clarify how body weight was measured: were participants instructed to wear minimal clothing, or was weight recorded in whatever clothing they were wearing on the day? Were they informed in advance to wear light clothing for accuracy?
- Indicate whether participants were asked to remove shoes prior to height measurement.
- For the PAQ-A (lines 215–219), incorporate the content into a cohesive paragraph rather than presenting it as bullet points.
Results
- In Table 2, indicate whether all groups (EFS, SPM, and KMS) are significantly different from the non-athlete group. If so, add symbols or p-values to the table to highlight these differences. Consider converting this table into a graph for clearer visualization of differences between athletes and non-athletes, while still describing the raw data in the text.
- In Table 3, what does (n/%) mean? Where are these numbers coming from? Please explain. Include, perhaps in the table legend, an explanation of what these values mean and their source.
- As with Table 2, note whether significant differences exist between EFS, SPM, and KMS compared to non-athletes in Table 3.
- Table 6 can also be converted to a graph, to show a visual representation of athletes vs non-athletes.
- Section 3.2, don’t need the first paragraph, can move that to methods if required. Report data only.
- Section 3.4, the first paragraph is more suited for the method section
Several portions of the Results section are interpretive in nature and would be more appropriately placed in the Discussion or the Methods section. The Results should focus exclusively on reporting data, with all interpretation and speculation reserved for the Discussion. For instance, lines 358–367 are better suited for the Discussion. Overall, this section requires the most substantial revision to ensure clarity and accuracy in presentation.
Discussion
- Ideally, the first paragraph of a discussion should briefly restate the study’s purpose and summarize the key findings in 2–3 sentences. This allows readers to immediately identify the most important results and understand their relevance. By establishing the main outcomes at the beginning, the paragraph sets the foundation for deeper interpretation, comparison with previous research, and exploration of broader implications in the subsequent discussion. Please address this.
- Section 4.1 should not open the Discussion, as it does not address the study’s central findings. Please restructure the section so that the main results are presented and discussed first. Section 4.1 can be moved later in the Discussion and shortened to maintain emphasis on the most critical outcomes.
In summary, this manuscript addresses an important and timely research question with a well-designed study and clear writing. However, substantial revisions are needed before it can be considered for publication. The most critical issues lie in the Results section, which currently includes interpretive statements that are better suited for the Discussion and methodological details that belong in the Methods section. The Discussion section also requires restructuring to highlight the main findings at the outset and to ensure a clear and focused interpretation of results. Finally, improvements in data presentation, such as clarifying tables, incorporating graphs, and reducing redundancy, would greatly enhance the clarity, readability, and overall impact of the manuscript.
Author Response
Dear Reviewer,
We would like to sincerely thank you for your careful analysis and constructive comments on our manuscript. We especially appreciate the fact that you highlighted both the strengths of the study (clarity of writing, relevance of the research question, adequacy of the design) and the aspects that need improvement. Your feedback has provided us with valuable guidance to refine the structure of the manuscript, clarify certain passages, and improve the presentation of the results.
We have taken all the comments into account and have revised the text accordingly. All additions and modifications made have been highlighted in blue in the manuscript, so that they can be easily followed. In addition, we have revised the entire text to correct the inconsistent use of short dashes, thereby improving the coherence and readability of the manuscript.
Your comment:
„Please mention examples of faculties with no advanced curricular components related to sports (non-sports group), so readers understand what the other population was studying (example: engineering, biology, etc).”
Response:
Thank you for this pertinent observation. We have completed the Participants section by adding concrete examples of faculties for the non-sports group (e.g., Engineering, Biology, Humanities, and Economics), to facilitate a better understanding of the study domains of this population. The modification is highlighted in lines 145–154.
Your comment:
„Please incorporate the bullet points of inclusion and exclusion criteria into a paragraph format. These can be separated into two paragraphs if needed. They both can be separate paragraphs if needed.”
Response:
Thank you for the recommendation. We have reformulated the section on inclusion and exclusion criteria, transforming the bullet-point list into a narrative format. The inclusion and exclusion criteria are now presented in a distinct paragraph. The modification can be consulted in lines 179–184.
Your comment:
„The sentence “To ensure the specificity and accuracy of responses, participants were instructed to refer exclusively to university-based social interactions involving other students, regardless of their faculty” is unclear. Please rephrase for clarity.”
Response:
Thank you for the observation. We have rephrased the sentence to increase clarity and avoid any ambiguity. In the revised version, the sentence now reads:
“To maximize contextual specificity, participants were instructed to report only experiences involving social interactions with fellow university students (excluding family, work, or non-university peers) when answering bullying/cyberbullying items.”
This modification can be consulted in lines 193–195.
Your comment:
„Clarify how body weight was measured: were participants instructed to wear minimal clothing, or was weight recorded in whatever clothing they were wearing on the day? Were they informed in advance to wear light clothing for accuracy?
Indicate whether participants were asked to remove shoes prior to height measurement.”
Response:
Thank you for these important observations. We have added further details in the Procedure and data collection instruments section to clarify how anthropometric measurements were conducted. Thus, we specified that:
- Body weight was measured with participants wearing light clothing, without requiring minimal clothing; participants were informed in advance to avoid heavy clothing for accuracy.
- Height was measured with participants barefoot, after removing footwear.
This addition was also fully in line with the observation made by Reviewer 2, which is why the modification appears highlighted in yellow in the manuscript (lines 201–207).
Your comment:
„For the PAQ-A (lines 215–219), incorporate the content into a cohesive paragraph rather than presenting it as bullet points.”
Response:
Thank you for the suggestion. We have revised the description of the PAQ-A questionnaire, transforming the bullet-point list into a coherent paragraph, to ensure a clearer flow of information and improved readability. The modification can be consulted in lines 222–225.
Your comment:
„In Table 2, indicate whether all groups (EFS, SPM, and KMS) are significantly different from the non-athlete group. If so, add symbols or p-values to the table to highlight these differences. Consider converting this table into a graph for clearer visualization of differences between athletes and non-athletes, while still describing the raw data in the text.”
Response:
Thank you for the observation. We have added asterisk (*) markers in Table 2 to highlight statistically significant differences between the athlete subgroups (EFS, SPM, KMS) and the non-athlete group, according to independent t-tests with Bonferroni correction (all p < 0.001). The modifications are highlighted in lines 338–341. Regarding the suggestion to convert the table into a graph, we preferred to keep the presentation in tabular format, as it provides a complete and detailed visualization of the raw values supporting the comparative analyses and subsequent discussion. We believe that the table, in this form, better serves the analytical objectives and methodological transparency of the study.
Your comment:
„In Table 3, what does (n/%) mean? Where are these numbers coming from? Please explain. Include, perhaps in the table legend, an explanation of what these values mean and their source.”
Response:
Thank you for the observation. We have added the following explanation in the footnote of Table 3: “n/%” denotes the count and the percentage within each program × sex subgroup; percentages are computed from the measured BMI using WHO cut-offs.” This clarification can be consulted in lines 352–353.
Your comment:
„As with Table 2, note whether significant differences exist between EFS, SPM, and KMS compared to non-athletes in Table 3.”
Response:
Thank you for the observation. We have clarified these differences and added the following note:
“Descriptive results by program and sex are presented in Table 3; asterisks (*) flag mean BMI values that differ significantly from the non-athlete group of the same sex (independent-samples t-tests, Bonferroni-adjusted within sex across the three athlete programs; all marked comparisons p < 0.001).”
The same explanation was also included in the footnote of Table 3. Both additions are highlighted in lines 345–348 and 353–355..
Your comment:
„Table 6 can also be converted to a graph, to show a visual representation of athletes vs non-athletes.”
Response:
Thank you for the suggestion. At the recommendation of Reviewer 2, Table 6 was revised and supplemented with additional information (t values, degrees of freedom, p-values, Hedges’ g, and r²). In this detailed form, we believe that the tabular presentation is more suitable than a graphical representation. The table allows readers to simultaneously view both mean differences and significance statistics, as well as effect sizes—elements that could not be fully illustrated in a graph without loss of information.
Your comment:
„Section 3.2, don’t need the first paragraph, can move that to methods if required. Report data only.”
Response:
Thank you for the observation. We have revised section 3.2, eliminating most of the first paragraph. We retained only the strictly necessary information that directly refers to the data in Table 2, to ensure coherence between the text and the presentation of results.
Your comment:
„Section 3.4, the first paragraph is more suited for the method section.”
Response:
Thank you for the observation. We have revised the first paragraph of section 3.4, in accordance both with your recommendation and with the earlier comments of Reviewer 2. In this way, section 3.4 remains focused exclusively on data presentation, while the methodological aspects are presented in the appropriate place. The modifications are highlighted in yellow in lines 359–366.
Your comment:
„Several portions of the Results section are interpretive in nature and would be more appropriately placed in the Discussion or the Methods section. The Results should focus exclusively on reporting data, with all interpretation and speculation reserved for the Discussion. For instance, lines 358–367 are better suited for the Discussion. Overall, this section requires the most substantial revision to ensure clarity and accuracy in presentation.”
Response:
Thank you for the well-argued observation and for the suggestions that guided the revision of this section. We have carried out a substantial restructuring of the Results section, removing interpretive paragraphs and retaining here only the objective reporting of data. The passages that contained interpretations or additional explanations, including the one corresponding to lines 358–367, were relocated to the Discussion or, where appropriate, to the Methods, where they fit more adequately and support the coherence of the text.
This restructuring clarified the presentation of results and ensured a stricter separation between data reporting and interpretation, which, in our view, has led to a significant improvement in the quality of the manuscript and increased overall readability.
Your comment:
„Ideally, the first paragraph of a discussion should briefly restate the study’s purpose and summarize the key findings in 2–3 sentences. This allows readers to immediately identify the most important results and understand their relevance. By establishing the main outcomes at the beginning, the paragraph sets the foundation for deeper interpretation, comparison with previous research, and exploration of broader implications in the subsequent discussion. Please address this.
Section 4.1 should not open the Discussion, as it does not address the study’s central findings. Please restructure the section so that the main results are presented and discussed first. Section 4.1 can be moved later in the Discussion and shortened to maintain emphasis on the most critical outcomes.”
Response:
Thank you for the very useful recommendation regarding the structure of the Discussion section. We have carefully followed your suggestions and made a consistent reorganization of this part. Thus, the interpretive paragraphs that were previously in the Results have been transferred to the corresponding sections of the Discussion, to respect the clear separation between data reporting and interpretation.
In the current form, the Discussion opens with an introductory paragraph that restates the study’s purpose and summarizes the main findings in a few concise sentences, allowing readers to immediately identify the central results and their relevance. This structure provides a solid foundation for subsequent comparisons with the specialized literature and for the exploration of theoretical and practical implications.
We believe that this restructuring has significantly improved the coherence and flow of the section, highlighting the essential findings from the beginning and ensuring a clearer, more focused, and better-integrated interpretation of the data obtained..
Your comment:
„In summary, this manuscript addresses an important and timely research question with a well-designed study and clear writing. However, substantial revisions are needed before it can be considered for publication. The most critical issues lie in the Results section, which currently includes interpretive statements that are better suited for the Discussion and methodological details that belong in the Methods section. The Discussion section also requires restructuring to highlight the main findings at the outset and to ensure a clear and focused interpretation of results. Finally, improvements in data presentation, such as clarifying tables, incorporating graphs, and reducing redundancy, would greatly enhance the clarity, readability, and overall impact of the manuscript.”
Response:
We sincerely thank you for your careful evaluation and for the positive appreciation of the relevance and overall quality of the study. We have rigorously revised the manuscript, taking into account all the recommendations made. The most important changes concern:
- The Results section, from which we removed interpretive passages and methodological details, relocating them to the Discussion and Methods, respectively, to ensure a clear separation between the objective reporting of data and their interpretation;
- The Discussion section, which was restructured to begin with the purpose and main findings, followed by in-depth interpretation and comparisons with the specialized literature;
- Data presentation, where we clarified the tables and added additional explanations in the footnotes, and in Tables 2 and 3 we introduced statistical markers to highlight significant differences.
We believe that these modifications directly address your observations and contribute to a clearer structure, a more focused text, and a more rigorous presentation of the results. We are confident that the revised version brings added coherence, readability, and scientific impact to the manuscript.
We especially appreciate your detailed and constructive observations, which guided us in the revision process and contributed essentially to improving the quality and scientific impact of the manuscript. We are grateful for the time and expertise you invested in evaluating our study.
The Authors

Round 2
Reviewer 3 Report
Comments and Suggestions for Authors
The authors have addressed all my questions, and the paper is now much improved. I only have a few minor comments:
Table 2: Please ensure consistency in decimal places. For example, change 2.20 to 2.2 to match the single decimal place format used throughout.
Section 3.3: Lines 345–348 are unnecessary since this information is already provided in the table description. These lines can be removed.
Section 4.5: The numerical data do not need to be included here, as the explanation of the underlying reasons is sufficient. I recommend removing the detailed numbers to streamline the discussion.
Author Response
Dear Reviewer,
We would like to sincerely thank you for your careful analysis and constructive comments on our manuscript. We are pleased to hear that you appreciated the improvements made. We have taken your observations into account, and the changes have been highlighted in purple for ease of reference.
Your comment:
„Table 2: Please ensure consistency in decimal places. For example, change 2.20 to 2.2 to match the single decimal place format used throughout.”
Response:
Thank you for this helpful observation. We have standardized all values in Table 2 to a single decimal place.
Your comment:
„Section 3.3: Lines 345–348 are unnecessary since this information is already provided in the table description. These lines can be removed.”
Response:
Thank you for pointing this out. We have removed those lines and kept only a concise reference to the Table 3.
Your comment:
„Section 4.5: The numerical data do not need to be included here, as the explanation of the underlying reasons is sufficient. I recommend removing the detailed numbers to streamline the discussion.”
Response:
Thank you for the suggestion. We have revised the paragraph in Section 4.5 to remove the numerical values (e.g., p-values, Hedges’ g, r², χ², ηp², B intervals) and retained only the qualitative and integrative interpretation.
We are grateful for the time and expertise you have invested in evaluating our study.
The Authors
